# Experimental genital tract infection demonstrates *Neisseria gonorrhoeae* MtrCDE efflux pump is not required for *in vivo* human infection and identifies gonococcal colonization bottleneck

Andreea Waltmann[1], Jacqueline T. Balthazar[2], Afrin A. Begum[3], Nancy Hua[4], Ann E. Jerse[3], William M. Shafer[2,5,6], Marcia M. Hobbs[1,7‡], Joseph A. Duncan[1,8,9‡*]

1 Division of Infectious Diseases, Department of Medicine, School of Medicine, University of North Carolina, Chapel Hill, North Carolina, United States, 2 Department of Microbiology and Immunology, Emory University School of Medicine, Atlanta, Georgia, United States, 3 Department of Microbiology and Immunology, Uniformed Services University, Bethesda, Maryland, United States, 4 The Emmes Company, Rockville, Maryland, United States, 5 The Emory Antibiotic Resistance Center, Emory University School of Medicine, Atlanta, Georgia, United States, 6 Laboratories of Bacterial Pathogenesis, Veterans Affairs Medical Center (Atlanta), Decatur, Georgia, United States, 7 Department of Microbiology and Immunology, University of North Carolina at Chapel Hill, Chapel Hill, North Carolina, United States, 8 Lineberger Comprehensive Cancer Center, University of North Carolina at Chapel Hill, Chapel Hill, North Carolina, United States, 9 Department of Pharmacology, School of Medicine, University of North Carolina, Chapel Hill, North Carolina, United States

‡ These authors are joint senior authors on this work.
* jaduncan@med.unc.edu

## Abstract

The MtrCDE efflux pump of *Neisseria gonorrhoeae* exports a wide range of antimicrobial compounds that the gonococcus encounters at mucosal surfaces during colonization and infection and is a known gonococcal virulence factor. Here, we evaluate the role of this efflux pump system in strain FA1090 during *in vivo* human male urethral infection with *N. gonorrhoeae* using a controlled human infection model. With the strategy of competitive infections initiated with mixtures of wild-type FA1090 and an isogenic mutant FA1090 strain that does not contain a functional MtrCDE pump, we found that the presence of the efflux pump is not required for an infection to be established in the human male urethra. This finding contrasts with previous studies of *in vivo* infection in the lower genital tract of female mice, which demonstrated that mutant gonococci of a different strain (FA19) lacking a functional MtrCDE pump had a significantly reduced fitness compared to their wild-type parental FA19 strain. To determine if these conflicting results are due to strain or human *vs.* mouse differences, we conducted a series of systematic competitive infections in female mice with the same FA1090 strains as in humans, and with FA19 strains, including mutants that do not assemble a functional MtrCDE efflux pump. Our results indicate the fitness advantage provided by the MtrCDE efflux pump during infection of mice is strain dependent. Owing to the equal fitness of the two FA1090 strains in men, our experiments also demonstrated the presence of

**Data Availability Statement:** All original data used in the analyses of the current manuscript are available in the supplementary tables.

**Funding:** The authors and study were supported by funding from the National Institutes of Health (NIH). Specifically, MMH and JAD were both supported through Grant Award Numbers U01AI114378 and U19AI113170 from the NIH National Institute of Allergy and Infectious Diseases. AW was supported by NIAID through Grant Award Number U01AI114378 and by the NIH the National Center for Advancing Translational Science (NCATS) though Grant Award Numbers TL1TR002491/UL1TR001111 and K12TR004416. AEJ was supported by NIAID through Grant Award Number U19AI113170. WMS was supported by NIAID through Grant Award Numbers U19AI113170 and R01AI021150. The content is solely the responsibility of the authors and does not necessarily represent the official views of the NIH. WMS is the recipient of a Senior Research Career Scientist Award from the Department of Veterans Affairs Medical Research Service. The funders had no role in study design, data collection and analysis, decision to publish, or preparation of the manuscript.

**Competing interests:** The authors have declared that no competing interests exist.

a colonization bottleneck of *N. gonorrhoeae* in the human male urethra, which may open a new area of inquiry into *N. gonorrhoeae* infection dynamics and control.

**TRIAL REGISTRATION.** Clinicaltrials.gov NCT03840811.

## Author summary

The work herein uses a unique controlled human infection model to examine the importance of a known *Neisseria gonorrhoeae* virulence factor, namely the multiple transferrable resistance (MtrCDE) efflux pump, to *in vivo* gonorrhea in human male urethra. The MtrCDE efflux pump is an important contributor to antimicrobial resistance because it enhances the ability of *N. gonorrhoeae* to withstand the action of multiple classes of antibiotics by facilitating the export of these compounds out of the bacterial cell. Antimicrobial resistant *Neisseria gonorrhoeae* is recognized by the US Centers for Disease Control as an urgent threat emerging in the antimicrobial resistance space and as a priority pathogen for antimicrobial resistance monitoring by the World Health Organization. Prior to the work described in this manuscript, studies performed *in vitro* and *in vivo* using the mouse model of infection had suggested that MtrCDE confers *N. gonorrhoeae* a within-host competitive advantage, even in the absence of antibiotic treatment, presumed to be related to the pump's ability to act on endogenous host derived antimicrobial factors. Our study demonstrates that the pump is not required for human infection and that it does not confer a competitive advantage in the human male urethral infection, at least in the context of the *N. gonorrhoeae* strain approved for use in experimental infections in humans. In addition, our study also revealed a previously unrecognized within-host bottleneck to the establishment of a productive urethral infection. Host bottlenecks have been described for other human pathogens, but never before for a bacterial sexually-transmitted pathogen. This observation opens an entirely new avenue of research inquiry that ultimately will help us better understand the pathogenesis of *N. gonorrhoeae* colonization and identify potential points for development of novel preventative measures.

## Introduction

*Neisseria gonorrhoeae*, the sexually-transmitted bacterium that causes gonorrhea, is exquisitely adapted to humans, its only natural host. The gonococcus is equipped with multiple mechanisms that allow it to quickly respond to host innate immune responses. The multiple transferrable resistance (Mtr) efflux pump is critical to the ability of *N. gonorrhoeae* to export a wide range of substrates that bathe mucosal surfaces, including host-derived antimicrobials (antimicrobial peptides, bile salts, progesterone, fatty acids, and cathelicidins, such as human LL-37 or its murine homologue CRAMP-38) and antibiotics [1–7]. The MtrCDE efflux pump is a member of the resistance-nodulation-division family possessed by many Gram-negative bacteria [4,8]. The Mtr system was first identified in 1973 as providing decreased susceptibility of gonococci to structurally diverse hydrophobic antibiotics, detergents, and dyes [9] and was postulated to be due to decreased permeability of the bacterial outer membrane to these antimicrobials [10]. It has also been associated with enhanced resistance to killing by human neutrophils and neutrophil products [11]. The Mtr-mediated resistant phenotype to antimicrobials and antibiotics is due to the relief of transcriptional repression of the *mtrCDE*–encoded efflux pump operon [3,12–14].

The *mtrCDE* operon is transcriptionally regulated by *cis-* and *trans*-acting control processes [1,15,16]. MtrD is the inner membrane component of the pump and is connected to MtrE in the outer membrane through the periplasmic membrane fusion lipoprotein MtrC [2,7,8]. Efflux is dependent on energy supplied by the proton motive force transduced by MtrD [17,18]. Expression of *mtrCDE* in wild-type gonococci is subject to repression by MtrR [3,12,13] and, in the presence of an inducer, activation by MtrA [13,14]. Loss-of-function *mtrD* mutations significantly reduce the ability of gonococci to colonize the lower genital tract of female mice in experimental infection [19]. Higher levels of the pump, on the other hand, either through induction by MtrA, or de-repression through either loss of MtrR or through mutations in the MtrR DNA-binding region enhance resistance to antibiotics [5,20–23], host-derived antimicrobials [5], and *in vivo* fitness in the mouse infection model [24]. In strains that overexpress the pump, loss of MtrCDE can increase antibiotic susceptibility in otherwise clinically-resistant gonococci [25], including strain H041, which was isolated from the first reported extended-spectrum-cephalosporin resistant case of gonorrhea [26]. These findings support the idea that the MtrCDE efflux pump is important during *in vivo* infection [5,24] and components of the pump have been proposed as vaccine candidates [27–30]. To formally investigate the requirement of the pump during *in vivo* human infection, we leveraged our group's model of experimental human gonococcal infection. This model has proven to be a safe and efficient means of studying human urethral gonococcal infection without serious complications since 1991 (reviewed in [31,32]). It takes advantage of the self-limited nature of gonococcal urethral infection in men, and infection can be initiated in male participants through intraurethral inoculation [31,32]. With this model and the strategy of competitive infections with bacterial mixtures containing wild-type strain FA1090 and an isogenic FA1090 mutant that lacks MtrD, we sought to determine the importance of the MtrCDE efflux pump to *in vivo N. gonorrhoeae* infection in the male urethra. We hypothesized that mutations that abrogate the expression of the Mtr efflux pump, such as loss of MtrD, would affect *N. gonorrhoeae* infectivity in the male urethra. Second, if during competitive infections the wild-type strain and isogenic mutant proved to be equally fit in establishing an infection (a null hypothesis scenario), we sought to demonstrate for the first time the existence of a gonococcal infection bottleneck. Studying infection bottlenecks provides important information about host-pathogen interaction and transmission restriction, which in turn is useful in exploring new infection prevention strategies.

## Results

### Infectivity and clinical courses of infection in non-competitive infections

The *in vivo* infectivity of the wild-type FA1090 and FA1090Δ*mtrD* strains (Table 1) was assessed using single-strain infections. Results are summarized in Table 2. The evaluable population included participants who received a dose of *N. gonorrhoeae* within 1 $\text{Log}_{10}$ of the intended dose of one million organisms, below which the inoculum size is considered low at

Table 1. Bacterial strains used in this study.

| Strain | *mtrCDE* pump expression | Reference |
|---|---|---|
| FA1090 | Not inducible due to 11 bp deletion in coding region of activator *mtrA* | [34] |
| FA1090Δ*mtrD* | None | This study |
| FA19 | Inducible (wild-type *mtrA*) | [66] |
| FA19Δ*mtrD* | None | This study |
| FA19 *mtrD::kan* | None | This study and [19] |

**Table 2. Clinical course parameters of non-competitive infections in men inoculated with *N. gonorrhoeae* FA1090 or *N. gonorrhoeae* FA1090Δ*mtrD*.**

| Participant | Strain | Inoculum size[a] | Infectivity (% infected, 95% confidence intervals) | Bacteriuria[b] | Pyuria[c] | Day of Treatment[d] |
|---|---|---|---|---|---|---|
| 232 | FA1090 | 6.6 | Y | 6.4 | 6.7 | 3 |
| 241 | | 6.8 | Y | 4.5 | 6.3 | 2 |
| 242 | | 6.7 | Y | 6.3 | 7.0 | 3 |
| 243 | | 6.8 | Y | 5.5 | 7.1 | 3 |
| 251 | | 5.9 | Y | 2.0 | 6.4 | 1 |
| 252 | | 6 | Y | 6.0 | 6.9 | 4 |
| *Summary[e]* | | *6.5 (6.0–6.9)* | *100% (54%, 100%)* | *4.8 (3.0–7.6)* | *6.7 (6.4–7.1)* | *3.0 (2.0–3.0)* |
| 216 | FA1090Δ*mtrD* | 5.8 | Y | 6.2 | 6.2 | 3 |
| 220 | | 6.4 | Y | 3.2 | 6.8 | 1 |
| 222 | | 6.7 | Y | 4.0 | 8.0 | 2 |
| 223 | | 6.7 | Y | 6.0 | 8.0 | 3 |
| 271 | | 6.0 | Y | 0.7 | 6.0 | 4 |
| 272 | | 6.1 | N | ND | 6.0 | 5 |
| 273 | | 6.1 | Y | 2.0 | 6.9 | 4 |
| *Summary[e]* | | *6.2 (5.9–6.6)* | *86% (42%, 100%)* | *3.0 (1.2–7.0)* | *6.8 (6.1–7.6)* | *3.0 (2.0–4.0)* |
| *p-value[f]* | | *0.39* | *0.54* | *0.23* | *0.77* | *0.50* |

Y = yes, N = no, ND = not detected

[a]$Log_{10}$ cfu *N. gonorrhoeae* mixture delivered intraurethrally

[b]$Log_{10}$ cfu *N. gonorrhoeae* recovered/mL urine sediment on day of treatment

[c]$Log_{10}$ white blood cells (wbc)/mL urine sediment on day of treatment

[d]Days from inoculation to antibiotic

[e]Summary statistics refer to geometric means with 95% confidence intervals shown in brackets, except for the infectivity summary, which denotes the percentage of participants deemed to have been infected (with 95% confidence intervals), and time to treatment, which denotes the median time from inoculation to treatment (with interquartile range).

[f]P-values were generated using the two-sided two-sample Wilcoxon rank-sum test, except for the infectivity comparison for which the one-sided Fisher's exact test was used.

approximately $ID_{50}$, and reached an objective study endpoint (urethral discharge or day 5). Six male participants were inoculated with wild-type FA1090 alone and included in downstream analyses. Of nine men inoculated with FA1090Δ*mtrD* alone, two were excluded: one volunteer received a low inoculum dose at approximately $ID_{50}$ [inoculum size = $Log_{10}(5.0)$] and a second volunteer requested treatment without signs, symptoms, or positive cultures on day 2 post inoculation. Among participants included in final analyses, there were no significant differences between the two strains with respect to infectivity, inoculum size, pyuria on treatment day, and time to treatment (Table 2). Among evaluable and infected volunteers, bacteriuria on treatment day did not differ significantly between the two strains (Table 2). No differences in *in vitro* growth rates were observed when wild-type FA1090 and FA1090Δ*mtrD* mutant bacteria were grown as pure strain side-by-side liquid cultures, as evidenced by optical density measurements (S2 Fig).

### The MtrCDE efflux pump does not confer a fitness advantage during urethral infection of men experimentally infected with strain FA1090 of *N. gonorrhoeae*

To determine whether the FA1090 MtrCDE efflux pump confers a fitness advantage to *N. gonorrhoeae* in the male urethra, we conducted competitive infections using mixed FA1090 and FA1090Δ*mtrD* inocula. Twelve participants in four cohorts were inoculated with strain

mixtures. The evaluable population for competitive index calculations included participants who received a dose of *N. gonorrhoeae* within 1 $Log_{10}$ of the intended dose of 1 million organisms and reached an objective study endpoint (urethral discharge or day 5) for whom a competitive index (CI) could be calculated. Of 12 participants, 10 were evaluable and included in competitive infection evaluations (Table 3). One volunteer remained urine culture-negative until the last day of post-inoculation follow-up when they were treated with antibiotics, as per protocol. A second volunteer did not comply with specimen collection requirements of providing first void urine. Among the evaluable participants, infectivity and course of infection parameters were in line with those in non-competitive infections (Table 3).

The proportion of each strain in the inoculum and in urine sediment from daily first-void urine was determined by sub-culturing up to 96 colonies from each positive urine sediment culture and identifying strain-specific colony forming units (cfu) by real-time colony PCR (S2 Table). When fewer than 96 colonies were present on the culture plate, all available cfu were picked, sub-cultured, and screened by real-time colony PCR. The ratio of mutant cfu relative to wild-type cfu was identified for each participant daily, including treatment day. The results were expressed as $Log_{10}$ competitive indices ($Log_{10}$ [CI]). The daily $Log_{10}$ (CI) are shown in Fig 1A. Within 24 hours after inoculation, 8/10 men (80.0%) were culture positive (CIs could not be calculated for the 2/10 participants with negative cultures, 20.0%), with only 1 strain recovered from 5/8 culture positive men (62.5%) and both strains recovered from 3/8 men (37.5%). Overall, on day 1 after challenge 4 outcomes favored the mutant and 4 outcomes favored the wild-type. Within 48 hours after inoculation, all 10 men (100.0%) were culture

**Table 3. Infectivity and course of infection parameters in men infected competitively with FA1090+FA1090Δ*mtrD*, and comparisons to the same parameters from non-competitive infections in men.**

| Participant | Strain | Inoculum size[a] | Infectivity (% infected, 95% confidence intervals) | Bacteriuria[b] | Pyuria[c] | Day of Treatment[d] |
|---|---|---|---|---|---|---|
| 281 | FA1090+FA1090Δ*mtrD* | 6.6 | Y | 2.9 | 7.1 | 2 |
| 291 | | 5.8 | Y | 5.2 | 6.9 | 4 |
| 292 | | 5.8 | Y | 5.0 | 6.5 | 4 |
| 300 | | 6.5 | Y | 4.6 | 6.9 | 4 |
| 301 | | 6.5 | Y | 5.1 | 6.5 | 3 |
| 302 | | 6.6 | Y | 5.0 | 6.3 | 2 |
| 303 | | 6.5 | Y | 5.0 | 6.6 | 4 |
| 311 | | 6.5 | Y | 3.2 | 6.9 | 2 |
| 313 | | 6.3 | Y | 3.5 | 6.8 | 4 |
| 314 | | 6.4 | Y | 3.5 | 7.0 | 4 |
| *Summary*[e] | FA1090+FA1090Δ*mtrD* | *6.3 (6.1–6.6)* | *100% (69%, 100%)* | *4.2 (3.6–4.9)* | *6.7 (6.6–6.9)* | *4 (2.0–4.0)* |
| *Summary* | FA1090 | *6.5 (6.0–6.9)* | *100% (54%, 100%)* | *4.8 (3.0–7.6)* | *6.7 (6.4–7.1)* | *3.0 (2.0–3.0)* |
| *Summary* | FA1090Δ*mtrD* | *6.2 (5.9–6.6)* | *86% (42%, 100%)* | *3.0 (1.2–7.0)* | *6.8 (6.1–7.6)* | *4.0 (4.0–4.0)* |
| *p-value*[f] | | *0.38* | *0.56* | *0.25* | *0.91* | *0.03* |

Y = yes, N = no

[a]$Log_{10}$ cfu *N. gonorrhoeae* mixture delivered intraurethrally

[b]$Log_{10}$ cfu *N. gonorrhoeae* recovered/mL urine sediment on day of treatment

[c]$Log_{10}$ wbc/mL urine sediment on day of treatment

[d]Days from inoculation to antibiotic treatment

[e]Summary statistics refer to geometric means with 95% confidence intervals shown in brackets, except for the infectivity summary, which denotes the percentage of participants deemed to have been infected (with 95% confidence intervals) and time to treatment, which denotes median time from inoculation to treatment (with interquartile range).

[f]P-values were generated using Kruskal-Wallis tests, except for the infectivity comparison for which the Fisher's exact test was used.

positive for *N. gonorrhoeae* and 4 outcomes favored the mutant and 5 outcomes favored the wild-type; the 10th participant was culture positive on this day, but fewer than 10 cfu were recovered (the lower limit of strain quantification), precluding the calculation of a CI for this participant on this day. For three of the 10 infected men (30.0%), the infection became symptomatic (clinically-observed urethritis) on day 2, which triggers antibiotic treatment. One of the 10 infected men (10.0%) became symptomatic and was treated on day 3, and 6/10 men (60.0%) became symptomatic and were treated on day 4.

The fitness of the FA1090Δ*mtrD* mutant relative to wild-type FA1090 was evaluated by computing the $Log_{10}$ (CIs) on the day of clinical urethritis when antibiotic treatment was given. The assumption was that by treatment day when clinical urethritis was apparent, the "winning" strain with a fitness competitive advantage had successfully established a site of infection by outcompeting the "losing" or less fit strain. $Log_{10}$ (CIs) were compared using a Wilcoxon one-sample signed rank test to a hypothetical median of 0. The null hypothesis was the absence of a competitive advantage for either strain. We observed equal outcomes favoring the mutant strain and the wild-type strain (Fig 1A and S2 Table) and thus the null hypothesis could not be rejected ($p = 0.477$, one-sample signed rank test). These findings demonstrate that expression of the FA1090 MtrCDE efflux pump does not confer a competitive advantage during *in vivo* experimental human urethral infection.

## The *in vivo* fitness advantage of the *N. gonorrhoeae* MtrCDE efflux pump is strain-dependent in female mice

Given that previous studies with FA19 *mtr* mutant strains in female mice showed that *mtrD* and *mtrE* mutants were attenuated for murine vaginal infection, and that the MtrCDE efflux pump is important for *in vivo* survival, either through induction by MtrA, the activator of the pump expression, or de-repression through loss of MtrR, the repressor of pump expression [5,19,24], the results of the human competitive infections with FA1090 and FA1090Δ*mtrD* were unexpected. To determine whether this discrepancy reflected differences between model systems, between *N. gonorrhoeae* strains, or between mutant generation methodologies, we conducted competitive infections in female mice with the following strain mixtures of wild-type and mutant: wild-type FA1090 with FA1090Δ*mtrD*, wild-type FA19 with FA19Δ*mtrD*, and wild-type FA19 with FA19*mtrD::kan*, which was tested previously [19]. For this study, groups of female BALB/c mice were vaginally inoculated with mixtures containing similar numbers of wild-type and mutant bacteria (total dose, $Log_{10}$ 5.8–6.0). Mouse vaginal swabs for quantitative *N. gonorrhoeae* culture were collected on day 1 post-inoculation and then every other day until the mice cleared the infection. All mice cleared the infection by day 7 post-inoculation. For mice competitively infected with FA1090 and FA1090Δ*mtrD* mixtures and with FA19 and FA19Δ*mtr* mixtures, single colonies were strain typed by real-time colony PCR. Quantitative selective culture was used for FA19 and FA19 *mtrD::kan* competitive infections to determine strain ratios.

Among the 11 mice competitively infected with FA1090 and FA1090Δ*mtrD* (Fig 2B and S3 Table), we observed five outcomes that favored the wild-type FA1090 and six outcomes that favored FA1090Δ*mtrD* ($p = 0.672$ 1-sided Fisher's exact, Fig 1B) on the final day of positive cultures. There was no significant departure from the null hypothesis, as the Log10 (CIs) were not from different from the hypothetical median of 0 ($p = 0.943$, Wilcoxon one-sample signed rank test). Daily CIs were available for one of the two cohorts of mice FA1090 and FA1090Δ*mtrD* (7 of the 11 mice, mice 5–11), which show the same equal fitness trend as CIs from the last day of positive cultures of the same subset of mice and of all 11 mice (Fig 2B and S4 Table). This result recapitulates the results of the human competitive infections, indicating

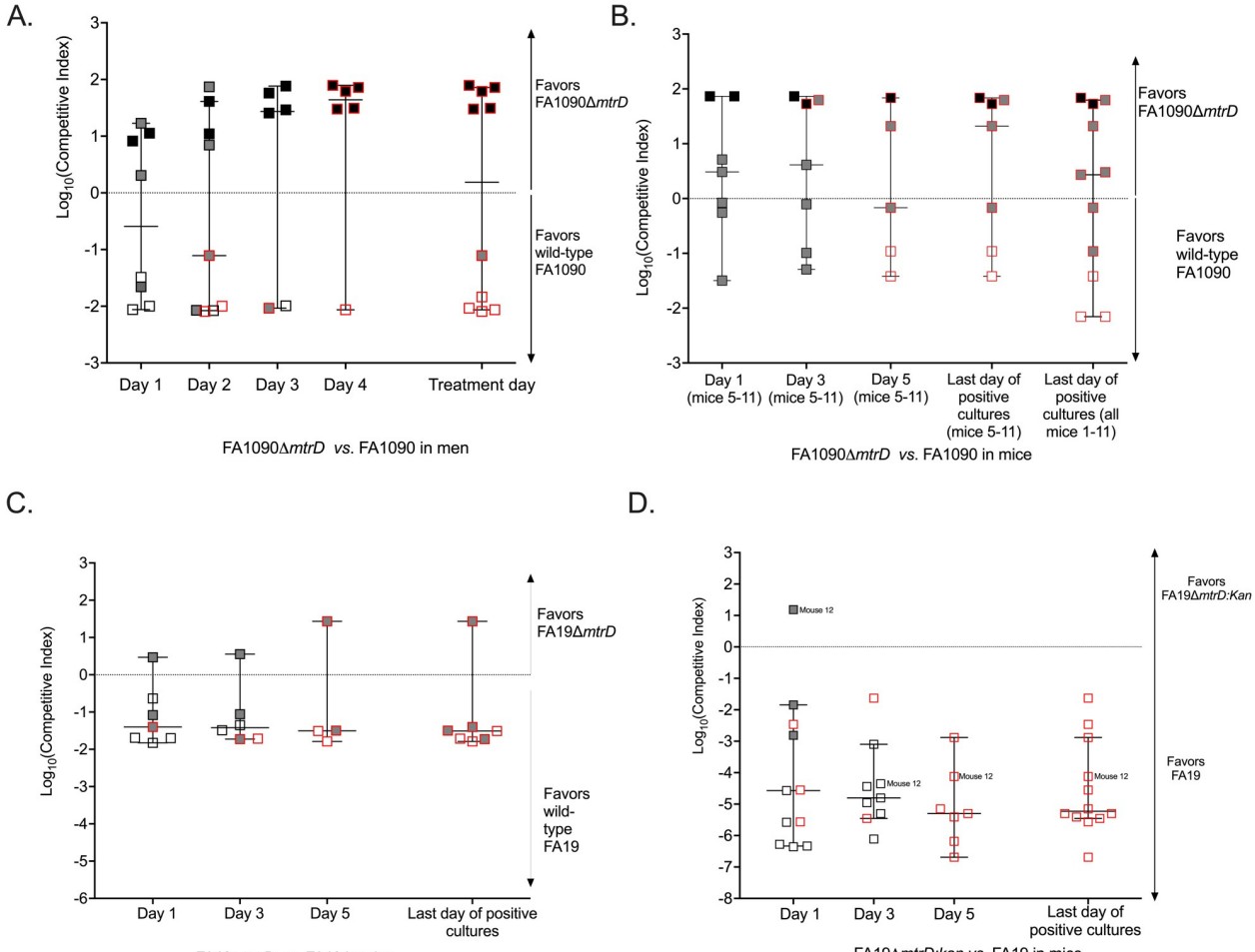

**Fig 1. Results of competitive infections in men and mice.** Panel A shows competitive infections of men with mixtures of FA1090 and FA1090Δ*mtrD*. Twelve men were competitively inoculated with mixtures of FA1090 and FA1090Δ*mtrD*. Participants were observed for signs and symptoms of clinical urethritis up to 5 days post-inoculation and received antibiotics as soon as urethral discharge developed or on the last day of study follow-up (day 5 post inoculation). First void urines were obtained daily after inoculation until the final study day and quantitatively cultured to determine infectivity. Inocula were also quantitively cultured. Ten men became infected and developed gonococcal urethritis with urethral discharge and their data were included in final analyses presented here. Colonies from the quantitative cultures of inocula and daily urine specimens were enumerated and up to 96 single colony isolates per culture per day were picked and stored until strain determination by real-time PCR. The colony real-time PCR results from inocula and daily urine were used to calculate daily competitive indices using the formula: CI = mutant cfu (output)/wild-type cfu (output) ÷ mutant cfu (input)/wild-type cfu (input). Output refers to the number of wild-type and mutant cfu enumerated from cultures of urine sediment or from cultures of mouse vaginal swab suspension. Input refers to the number of cfu enumerated from culture of bacterial suspension used to inoculate that cohort/group of men or mice. Thus, CIs reported for each participant and mouse were calculated with the exact proportion of strains identified in the inoculum of the group/cohort they were a part of. CIs are graphed on the logarithmic scale. Log$_{10}$CI greater than 0 indicated the mutant was favored and Log$_{10}$CI less than 0 indicates that the wild-type was favored. Horizontal bars represent median CIs. White squares refer to men from whom only the wild-type was recovered. Grey squares denote men from whom a mix of the two strains was recovered. Black squares denote men from whom only the mutant strain was recovered. White, grey, or black squares with red borders refer to the treatment day results when the participant became symptomatic (clinical urethritis with urethral discharge) and/or was treated with antibiotics because it was the day of post-inoculation observation (day 5). Panels B, C, and D show the results of the mouse competitive infections. Eleven mice were inoculated with mixtures of FA1090 and FA1090Δ*mtrD* (B), 7 mice were inoculated with mixtures of FA19 and FA19Δ*mtrD* (C), and 6 mice were inoculated with mixtures of FA19 and FA19*mtrD*::kan (D). Mouse CIs were calculated using the same formula as for the men and are also graphed on the logarithmic scale with horizontal bars denoting the median CIs. For mice inoculated with mixtures containing deletion strains (B and C), a total of 48 single colonies recovered from mouse vaginal swabs collected on day 1 after inoculation and then every other day (day 1, day 3, and day 5) until the last day of positive cultures (all mice cleared the infection by day 7; last day of positive cultures for all mice was day 5) were picked and individually screen by real-time colony PCR to determine the strain for each colony. For competitive infections with FA19 and FA19*mtrD*::kan, strain determination was done with quantitative selective culture by culturing inocula and vaginal swab suspensions on GC agar containing streptomycin (to recover and enumerate total gonococci) and GC agar containing streptomycin and kanamycin (to recover and enumerate mutant gonococci). We divided the number of kanamycin resistant cfu by the total number of gonococci recovered. Longitudinal strain composition data are available from all mouse competitive infections, except for 1 of the 2 cohorts of mice inoculated with FA1090 and FA1090Δ*mtrD* (n = 4 mice, mice 1–4, Panel B), for whom only 96 single colonies from the last day of positive cultures were saved

for real-time colony PCR screening; for the remaining 7 mice inoculated with FA1090 and FA1090Δ*mtrD* (mice 5–11) day 1, day 3, and day 5 colony real-time PCR data are available. White squares refer to mice from whom only the wild-type was recovered on last day of positive cultures. Grey squares denote mice from whom a mix of the two strains was recovered. Black squares denote mice from whom only the mutant strain was recovered. White, grey, or black squares with red borders refer to the results from the final day of positive cultures before the mouse cleared the bacteria. Note the different y axis scale for Panel D, imposed by the lower limit of detection of quantitative selective culture on GC agar for the FA19 +FA19 *mtrD*::kan competitive infections.

that FA1090Δ*mtrD* does not have a fitness defect during *in vivo* infection of men and mice relative to the parental wild-type strain. The lack of a fitness defect was also noted when FA1090 and FA1090Δ*mtrD* were grown competitively *in vitro* in liquid cultures (S5 Table). One difference between the results of the human and mouse competitive infections with FA1090 and

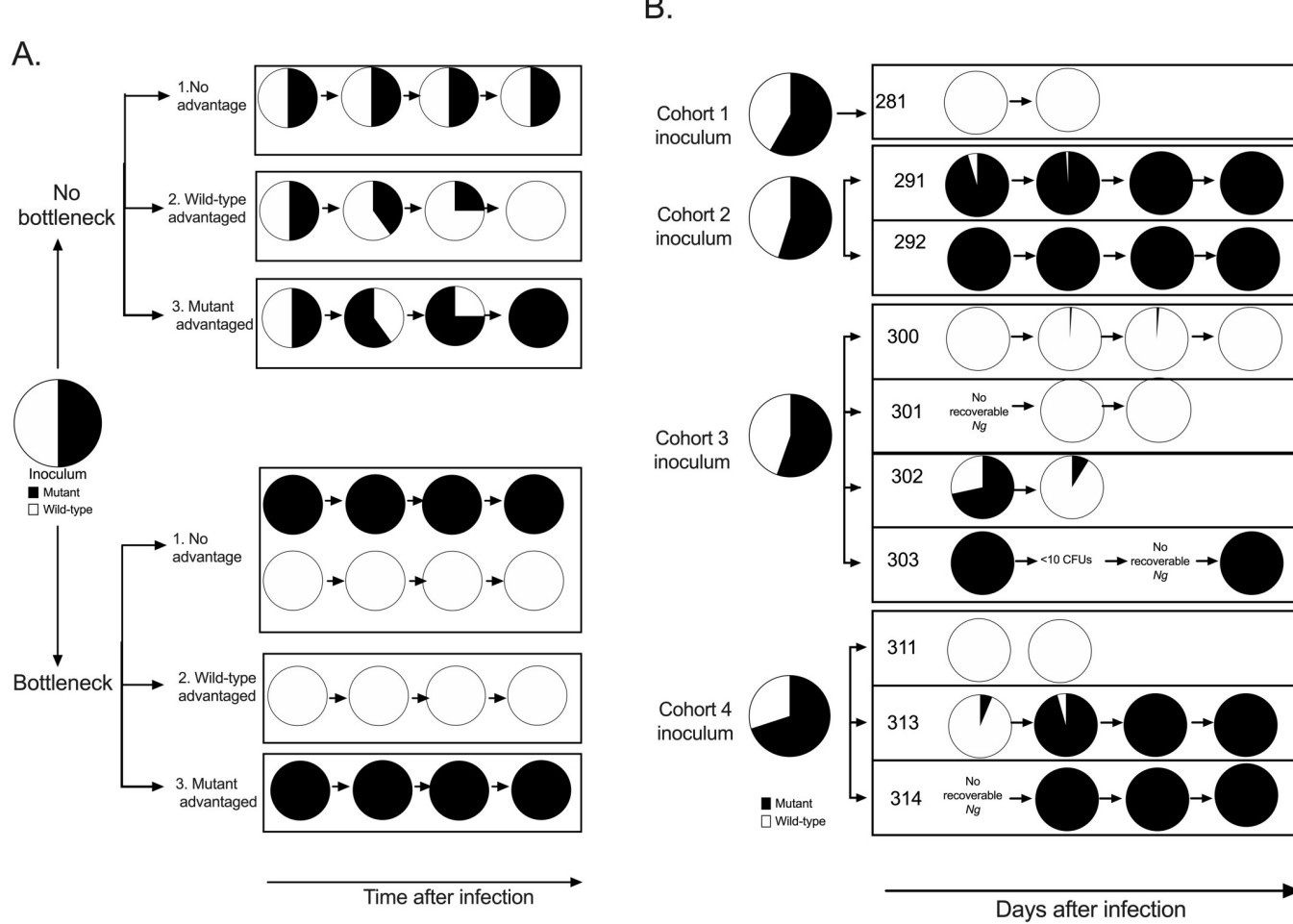

**Fig 2. Gonococcal strain dynamics during the early course of infection.** Panel A. Theoretical infection outcomes with respect to strain composition for infections initiated from 50:50 mixtures of two strains, shown in black and white, that differ in their competitive advantage in scenarios of bottleneck present early in the infection or no bottleneck (adapted with permission from [31]. Each pie chart represents a gonococcal population containing predicted proportions of the two theoretical strains. When the two strains have equal fitness and a bottleneck is present early in the course of infection that results in a population size restriction, we predict that equal numbers of outcomes favoring each strain will be observed. Panel B. Empirically observed outcomes in the current study for infections initiated with mixed inocula. Two to four men were inoculated with the same inoculum preparation in each of the four study cohorts. A total of 10 evaluable participants were available for final analysis (281, 291, 292, 300, 301, 302, 303, 311, 313, 314). The pie chart representing the average inoculum from the four cohorts refers to the mean strain composition of the four independent inocula. Each participant pie chart represents the daily mixture of gonococcal strains recovered from the urine over the 1- to 5-day period after inoculation. Last pie chart in the series for each participant refers to the study treatment day when clinical urethritis was apparent and/or when antibiotic treatment was administered.

FA1090Δ*mtrD* was that more mice than men had mixed strain recovery on the final positive culture day (mice: 6 /11, 54.5%; men: 1/10, 10.0%, *p* = 0.045 one-sided Fisher's exact).

Among the 7 mice infected with mixtures of FA19 and FA19Δ*mtrD*, all but one outcome on the final culture positive day favored wild-type FA19, with a significant difference from the hypothetical median of 0 (*p* = 0.031, Wilcoxon one-sample signed rank test, Fig 1C). The daily CIs for these mice are in line with what was observed on the final day of positive cultures (Fig 1C and S4 Table). Unlike the FA1090Δ*mtrD* mutant whose relative fitness to its parental FA1090 strain is not impaired during mouse vaginal colonization, the FA19 mutant that lacks a functional MtrCDE efflux pump is at a competitive disadvantage relative to wild-type FA19 in the female mouse. The FA1090 and FA19 results were significantly different in the mouse model (*p* = 0.047, Mann-Whitney rank sum test). All outcomes from the final day of positive cultures in the 6 mice competitively infected with FA19 and FA19 *mtrD*::kan favored the wild-type, with no mixed strain recovery (*p* = 0.030, Wilcoxon one-sample signed rank test, Fig 1D and S6 Table) in line with previously published results [5,19,24].

We conclude that the *in vivo* phenotype of the FA1090 mutant that does not produce the MtrCDE pump is the same in the male urethritis and female mouse models, but that an interesting strain difference between FA1090 and FA19 mutants was detected in the mouse model.

## Strain dynamics during human infection indicate the presence of a colonization bottleneck

Lack of a functional FA1090 MtrCDE efflux pump did not alter the fitness of the isogenic mutant relative to wild-type FA1090. Thus, our strategy of competitive infection enabled the incidental observation of a urethral colonization bottleneck. Competitive infections with strains of equal fitness have been used to study bottlenecks in other infection models. A bottleneck refers to barriers to colonization or infection that substantially reduce the original population size in an inoculum, such that a lower number of surviving organisms establish colonization and continue replicating in the host. Our findings indicate that such a bottleneck exists in the human male urethra early in the course of gonococcal infection.

Fig 2A depicts the theoretical outcomes of one individual infection initiated with a 50:50 mixture of two hypothetical strains under conditions of bottleneck and no bottleneck. When a bottleneck is not present, the strategy of competitive infections can have two outcomes: if both strains have equal fitness, the bacteria recovered from the host after inoculation mirrors the inoculum (outcome 1). If one of the strains has an advantage, the expectation is for that strain to progressively take over the dominate over the gonococcal population over course of infection (outcome 2 or 3). Under the scenario of a bottleneck, when neither has a fitness advantage and a bottleneck is present, both strains will have equal chance of passing through the bottleneck and establishing an infection (outcome 1); therefore, equal numbers of competitive infections would be dominated by each strain. If a strain is at a competitive advantage, then it is expected to dominate from early until late infection (bottleneck outcome 2 or 3).

Fig 2B describes the empirically observed outcomes for urine culture positive days in each of the 10 evaluable human challenge participants inoculated in 4 cohorts. The strain composition of the 4 inocula are also shown. In most men (8/10, 80.0%, Fig 2B), the predominant FA1090 strain on treatment day began dominating early in the infection (day 1 post-inoculation). In 2/10 men (20.0%), the predominating FA1090 strain from treatment day was different than the strain dominating the recovered *N. gonorrhoeae* cfu from day 1 post-inoculation. In all but one man (9/10, 90.0%), the strain composition of the first void urine on the final study day was uniformly dominated by either strain. Strain dynamics from urine culture positive days support the gonococcal bottleneck hypothesis.

## Discussion

In the female mouse model of gonorrhea, the MtrCDE efflux pump was previously found to be important to *in vivo* infection with *N. gonorrhoeae* strain FA19 [5, 19, 24]. These pre-clinical results had solidified the status of the MtrCDE efflux pump as a promising vaccine target. We anticipated human challenge studies of the MtrCDE efflux pump to corroborate the preclinical findings. Contrary to our hypothesis, competitive infections with wild-type FA1090 and an isogenic mutant of FA1090 that lacks a functional MtrCDE efflux pump showed that the two strains had equal fitness.

The previous investigations using the murine model were conducted with wild-type strain FA19 and FA19 mutants whose MtrCDE efflux pump was rendered non-functional via insertional inactivation [5,19,24]. We sought to determine the source of the discrepancy between human and mouse results and three possible sources of variation were considered: differences between gonococcal strains (FA1090 *vs*. FA19), differences in models of infection (the human male urethra *vs*. the lower genital tract of the female mouse), and differences in mutagenesis methodology (isogenic clean deletion mutagenesis without a selectable marker *vs*. insertional mutagenesis with a selectable marker).

To separate potential differences with respect to strain, we constructed an isogenic, unmarked Δ*mtrD* mutant of FA19 and conducted competitive infections in mice with both FA1090 and FA19 wild-type parent strains and their corresponding mutants (FA1090Δ*mtrD* and FA19Δ*mtrD)*. Both FA1090Δ*mtrD* and FA19 Δ*mtrD* lack a functional MtrCDE efflux pump due to the clean unmarked deletion of the *mtrD* gene, which is genetically identical in the two strains. The FA1090 strains used in human and mouse infections were part of the same lot of FA1090, FA1090Δ*mtrD*, or mixed FA1090 and FA1090Δ*mtrD* bacteria. FA1090 is currently the only human challenge inoculum strain approved for use in human challenge studies. The outcomes of competitive infection with FA1090 strains did not differ between humans and mice, indicating that the presence of an MtrCDE pump is not required for colonization of the female mouse lower genital tract or the human male urethra. By contrast, mouse competitive infections with FA19 strains showed that FA19 wild-type had a significant competitive advantage relative to FA19Δ*mtrD*. This competitive advantage was maintained when tested against the insertionally inactivated FA19 *mtrD* mutant used in the previous report [19]. Though differences were noted between the infection models (e.g., mixtures of FA1090 and FA1090Δ*mtrD* were recovered more frequently from the same mouse than same male) and between mutagenesis strategies (e.g., mixtures of FA19 and FA19Δ*mtrD* gonococci were more frequently recovered from the same animal than mixtures of FA19 and FA19*mtrD*::kan gonococci), our findings indicate that the fitness advantage conferred by the MtrCDE efflux pump in the murine infection model is primarily dependent on *N. gonorrhoeae* strain. FA1090 naturally expresses reduced levels of the MtrCDE pump compared to FA19, and other strains, due to an 11 bp deletion in the coding region of *mtrA* that results in loss of MtrA-mediated gene activation and a premature truncation of this transcriptional activator [14,23]. Thus, it is likely that the null mutation we introduced in the *mtrD* gene of FA1090 was insufficient to enact fitness phenotype differences compared to the parental FA1090 strain that already expresses a low level of MtrCDE (*i.e*., low expression in the wild-type *vs*. no expression in the mutant).

The generalizability of strain-dependent requirement of the MtrCDE efflux pump to human *in vivo* infection remains to be established. This hypothesis cannot be formally tested at this time with human challenge studies, because FA1090 is currently the only gonococcal strain with regulatory approval by the Food and Drug Administration for use in investigations with the human model of gonorrhea infection. Thus, human experimental infection with FA19 strains were not possible to formally evaluate that the requirement of the MtrCDE pump

is strain dependent in the human male urethra. Nevertheless, a recent genomic epidemiological study has revealed that gonococcal strains harboring mutations that result in the down-regulation or loss of function of the MtrCDE efflux pump are common in nature [33]. The study investigated mutations in the *mtrCDE* operon and its regulatory genes *mtrA* and *mtrR* using almost 5,000 global *N. gonorrhoeae* genomes and found that approximately 1 *N. gonorrhoeae* isolate in every 13 and 1 in every 25 had non-inducible expression of the MtrCDE pump or loss-of-function mutations that likely resulted in the loss of pump activity altogether, respectively [33]. This suggests that gonococcal strains can adapt through reduction or loss of a functional MtrCDE efflux pump and still establish productive infections, consistent with our results here. Further, global *N. gonorrhoeae* strains with mutations that downregulate or abrogate MtrCDE efflux pump activity were overrepresented in cervical *vs.* urethral isolates [33], suggesting that our results may be generalizable to human female genital infection; FA1090 was originally isolated from the endocervix of a female with probable disseminated gonococcal infection [34]. This line of inquiry cannot be formally pursued, however, because safety concerns of using the controlled human model of infection in females (i.e., risk of ascending infection) limits the use of the model to male urethral infections. Similarly, the generalizability of our results with the human model to extragenital sites of infection (rectal and oropharynx) is unclear because controlled models of gonorrhea infection do not exist for these anatomical sites of infection. Nevertheless, the global genomic analysis did identify naturally occurring *N. gonorrhoeae* strains with downregulated or likely inactive MtrCDE efflux pumps in a minority of pharyngeal and rectal isolates, though scarcity of genomes from these anatomical sites precludes accurate estimation of how common these variants are in extragenital infection.

Combined, our data and those of the global genomic analyses of mutations in the *mtrCDE* operon and regulatory regions, demonstrate it is unlikely that a vaccine targeting MtrCDE will prevent infection acquisition, at least in the male urethra. It is possible that antibody against MtrCDE could still contribute to protection through targeting of *N. gonorrhoeae* for antibody mediated killing or by downregulating or inactivating MtrCDE efflux pump activity. Host immune mediated inactivation or downregulation of MtrCDE could render the bacteria more susceptible to antimicrobial substances, including antibiotics, and promote natural clearance of gonococci after infection was established. The idea of an MtrCDE-based vaccine as adjunctive therapy to antibiotics is supported by evidence that that loss of MtrCDE function in clinically-resistant gonococci that overexpress the pump increases their antibiotic susceptibility [25]. The emergence of drug resistance and the dwindling arsenal of approved antibiotics that can effectively treat gonorrhea highlights the need for alternative gonorrhea control strategies.

The equal fitness of FA1090 and FA1090Δ*mtrD* during *in vivo* competitive infection of male urethras enabled the identification of a within-host colonization bottleneck. To our knowledge, this is the first demonstration that productive infection with *N. gonorrhoeae* is likely initiated by a restricted number of gonococci rather than through a uniform expansion of the initial mixed inoculum, though the concept of a gonococcal colonization bottleneck was previously proposed by our group [31]. Infection bottlenecks are drastic reductions in population size followed by population expansion of the "founder" organisms that initiate the productive infection. Bottlenecks have been observed in enteric and systemic human infections caused by pathogenic bacteria [35–41]. Among sexually-transmitted infections, human immunodeficiency virus 1 (HIV-1) bottlenecks have been described [42–45], but to date no studies have demonstrated the presence of bottlenecks in humans for bacterial sexually-transmitted infections. A study of uropathogenic *Escherichia coli* using a mouse model of urinary tract infection and single transposon mutagenesis with whole genome coverage of 2,913 unique nonessential genes showed that only 36 of these affected colonization fitness, which allowed a significant number of mutants to establish an infection [46]. Similarly, a murine neisserial

commensal study of oropharyngeal and gut colonization did not find evidence for a bottleneck at these anatomical sites after inoculation with a transposon library covering the genome [47]. However, a monoclonal antibody targeting LOS carbohydrates was shown to drive a murine bottleneck for invasive pathogenic *Neisseria meningitidis* [48], the only other pathogenic species in the *Neisseria* genus that shares 80–90% of its genome with the gonococcus, which typically inhabits the upper respiratory tract.

While our human challenge study was not designed *a priori* to investigate bottlenecks during *in vivo N. gonorrhoeae* infection, we can use our findings to hypothesize what the timing and size (the number of organisms able to breach the initial barrier) of the gonococcal bottleneck in the human male urethra might be.

First, we tracked the daily dynamics of the two infecting strains during human competitive infections from day 1 post-inoculation until the day of evident clinical urethritis when the infection became symptomatic with urethral discharge, up to 5 days post-inoculation; on average, participants developed urethritis 3 days after inoculation. We noted that the "winning" strain outcompeted the "losing" strain within the first 24h of inoculation in the majority of participants (8/10) and persisted as the dominant strain until the final study day. Interestingly, in 2 participants (302 and 313) in which both strains were recovered, the strain dominance observed on the first day after inoculation reversed the following day. Based on these data, we propose that gonococci encounter the bottleneck 24–48 hours after inoculation. The timing of our urine sampling (every 24 hours) is too infrequent to narrow down the timing of the bottleneck any further. Previous studies conducted by our group on the kinetics of opacity protein selection and pilin antigenic variation during experimental human infection initiated with FA1090 gonococci with defined antigenic repertoires of Opa and pilin support the idea that the infecting population encounters strong selective forces early in the infection process [49–52]. In the first 24 hours post-inoculation considerable Opa and pilin variation relative to the inoculum is observed during *in vivo* human infection [49–52], which are not observed when inoculum gonococci, without passage through a human, are cultured for 24 hours *in vitro* [49,52].

Second, both inoculating strains were recovered from 8/29 positive cultures of 4/10 volunteers. The recovery of both strains from the same clinical specimen during incipient infection indicates that the size of the bottleneck in the human male urethra is wider than one single organism. The exact size cannot be inferred at this time through the study of only two strains with equal fitness. The only described bottleneck for a sexually-transmitted pathogen, HIV-1, is restrictive and 80% of all HIV-1 heterosexual transmission is initiated from a single viral genetic variant [42–45], reviewed in [41], though for human bacterial pathogens both single-cell and multi-cell bottlenecks have been described [37,39,53–57]. It is interesting that our studies using the murine model do not demonstrate a bottleneck similar to what was observed in the human urethral infection model. This limits the use of mouse studies to explore the biologic basis for the bottleneck in the human urethra and advance these initial findings; mice clear the infection within 5–7 days, while in human males experimentally infected with *N. gonorrhoeae*, symptoms and gonococcal burden is generally uptrending after infection is established. The lack of an obvious bottleneck in the mouse that restricts *N. gonorrhoeae* colonization could potentially be more reflective of infection dynamics during human female genital infection, but due to limitations noted above in the safety of the gonorrhea challenge model for human females, it is not possible to pursue this type of study.

Our current findings from the human competitive infections are unable to indicate with any level of certainty whether the bottleneck is likely to be selective or non-selective. A selective bottleneck occurs when only organisms of a defined genetically encoded phenotype can overcome the bottleneck. In a non-selective bottleneck, strains overcome the bottleneck with equal probability, as may be case for *Streptococcus pneumoniae* [37,54–57]. Mapping the genomic

and phase variation changes of daily recovered gonococci from human challenge participants relative to the inoculum variants with deep sequencing methods will provide additional resolution on whether the host bottleneck impacts the diversity of the gonococcal population by applying pressure on specific antigenic loci.

Finally, reductions in the genetic diversity of pathogen populations can impact their transmission fitness. Our controlled human model of infection is not developed to study transmission, and it is currently unknown whether the bottleneck we observed during human experimental urethral infection impacts transmission of gonococci. It is worth noting, however, that phase and antigenic variation have long been a defining characteristic of *N. gonorrhoeae* and the recovery of antigenically heterogenous gonococci from the urethra of experimentally infected males relative to the antigenic characteristics of inoculating gonococci [49,50,52], as described above, speaks to the possibility that the bottleneck is selective and *N. gonorrhoeae* deploys phase and antigenic variation in response to ensure a diverse population is fit for onward transmission to a new site of infection. Future studies aimed to reveal which host factors drive bottleneck or which genes are essential to the survival of *N. gonorrhoeae* under bottleneck conditions will begin providing the information needed to explore whether the bottleneck can be exploited to thwart *N. gonorrhoeae's* ability to establish an initial infection and transmit to the next site of infection.

Overall, our study is providing compelling new information regarding two aspects of gonococcal biology that are relevant to our quest to develop new anti-gonococcal control strategies. First, by showing that a strain of *N. gonorrhoeae* is able to initiate a productive infection in the human male urethra even when the MtrCDE efflux pump, a well-studied *N. gonorrhoeae* virulence factor, is not functional we have generated data that informs future gonorrhea vaccine and drug development efforts that target the function of the Mtr efflux pump. Second, we demonstrated for the first time for a bacterial sexually-transmitted pathogen that a bottleneck exists during *in vivo* human urethral infection. In doing so, we have pointed towards a gonococcal Achilles heel that could be exploited for the development of novel infection control strategies. Thus, the implications of our findings could have the potential of ushering in fresh lines of inquiry in our quest for new and effective ways to confront gonorrhea. This is important because in the face of increased incidence and spread of antibiotic-resistant *N. gonorrhoeae* strains, the need to contain the threat of untreatable gonorrhea has become paramount.

## Materials and methods

### Ethics statement

Written informed consent from each participant was obtained prior to conducting any study activities. Experimental human infections were conducted between 2017–2019 in the Clinical and Translational Research Center of the North Carolina Translational and Clinical Sciences Institute at the University of North Carolina at Chapel Hill according to the guidelines of the U.S. Department of Health and Human Services and with approval by the University of North Carolina at Chapel Hill Institutional Review Board (approved protocol 12–0482) under an investigational new drug (IND) authorization by the U.S. Food and Drug Administration (IND 15123). All animal experiments were conducted at the Uniformed Services University of the Health Sciences according to the guidelines of the Association for the Assessment and Accreditation of Laboratory Animal Care under a protocol that was approved by the University's Institutional Animal Care and Use Committee.

## Strains and lots of *Neisseria gonorrhoeae* strains

*N. gonorrhoeae* strains used in this study are listed in Table 1. *N. gonorrhoeae* FA1090 served as the parent strain for the human challenge investigations, whereas FA1090 and FA19 were the parent strains for the mouse challenge experiments. FA1090 is a porin serotype PIB-3, streptomycin (Sm)-resistant strain that has been used extensively in experimental human infection studies [31]. FA19 has been used widely in mouse challenge studies, including studies of the Mtr efflux pump. FA1090 is the only human challenge inoculum strain in continuous use in human challenge studies since the inception of the human model of gonorrhea. FA1090 A26 is the lot of wild-type FA1090 utilized in the current study. The FA1090 A26 lot was prepared and extensively characterized according to procedures described in a Chemistry Manufacturing and Controls (CMC) document submitted to the Food and Drug Administration (FDA). It is used in human challenge studies through an investigational new drug (IND) authorization by the FDA (IND 15123). The lot of FA1090Δ*mtrD* derives from FA1090 A26. To be used in human challenge studies, the same lot-release characterization procedures were followed, with the addition of genotype confirmation of the mtrD deletion (confirmed by PCR and whole genome sequencing), purity testing on LB agar, which does not support gonococcal growth but would identify the presence of contaminants, and PolymyxinB susceptibility testing. The lot of FA1090 A26 and of FA1090Δ*mtrD* mixture was prepared after the individual working banks for each strain were made, characterized, and lot-released. To prepare the mixture lot, approximately equal concentrations of FA1090 A26 and of FA1090Δ*mtrD* were combined, dispensed into single-use vials, and frozen. The contents of one frozen vial of mixture were then sub-cultured for quantitative enumeration of colony forming units (cfu) of each strain by PCR, which confirmed that the mixtures in the working bank had the desired composition of approximately 50:50 mutant:wild-type.

The three working banks used in the human challenge studies presented here (FA1090 A26, FA1090Δ*mtrD*, and FA1090 A26 mixed with FA1090Δ*mtrD*) contain frozen, single-use vials of viable gonococci from the same lot of bacterial stocks. Mice were challenged with the same lot-released stock of FA1090 A26 and FA1090Δ*mtrD* mixtures as used in human studies; in addition three other FA19 strains were used in mouse challenge experiments (FA19 wild-type and two mutant strains lacking a functional Mtr pump, namely FA19Δ*mtrD*, and FA19 *mtrD*::*kan*). To generate the inocula that is administered to humans (or mice), one vial of frozen, viable gonococci is thawed and sub-cultured onto GCB agar to generate the actively growing bacteria for the inoculum suspension, as described below in "*Human urethral experimental infections*" and "*Murine competitive infections*" methodology sections.

## Generation and characterization of the mutant strains

To delete the *mtrD* gene and generate FA1090Δ*mtrD*, a single colony of the FA1090 A26 as the starting material was used in a two-step transformation, as previously described [58] and used by us to delete *mtrD* in FA1090. The deletion was made without leaving behind a selectable marker (S1 Fig). Briefly, a two-gene cassette containing both a selectable marker (chloramphenicol acetyl transferase [CAT] conferring chloramphenicol [Cm] resistance) and a counter selectable marker (*rpsL*, conferring streptomycin [Sm] susceptibility on the naturally resistant FA1090) was cloned into the *mtrD* gene and used to replace the wild-type gene on the chromosome by allelic exchange. A second transformation replaced the cassette-containing version of the gene with an unmarked deletion. Targeted PCR amplification of the *mtrD* locus and whole-genome sequencing of the FA1090Δ*mtrD* genome sequence confirmed deletion of the gene in the mutant. FA19Δ*mtrD* was generated in the same manner as FA1090Δ*mtrD* (i.e., leaving behind a clean deletion without a selectable marker), whereas the FA19 *mtrD*::*kan*

mutant was constructed via an insertional mutation, leaving behind a kanamycin selectable marker, as previously described [19]. While the mutant strains express the other two components of the Mtr efflux pump, the lack of *mtrD* expression leads to the inability to assemble the pump. MtrD expression contributes to gonococcal resistance to cationic antimicrobial compounds such as polymyxin B (PB). Compared to the FA1090 parental strain the *mtrD* deletion mutant was four-fold more PB susceptible (MICs of 100 vs. 25 µg/ml, respectively). *In vitro*, FA1090 and FA1090Δ*mtrD* had equal growth rates when compared side by side in liquid cultures.

## Human urethral experimental infections

Procedures for human male participant recruitment, informed consent, intraurethral inoculation, an antibiotic treatment were as previously described [31,32,59,60]. Females are excluded from these studies due to risks of ascending gonorrhea infection into the upper reproductive tract. Human challenge studies were conducted at the University of North Carolina at Chapel Hill, North Carolina, between April 2017 and November 2018. Separate cohorts of up to 4 participants were inoculated with FA1090 A26 alone, FA1090Δ*mtrD* alone, or with mixtures of FA1090 A26 combined with FA1090Δ*mtrD*. Procedures pertaining to inoculum preparation for single and mixed strain infections have been previously described [31,32,59,61,62]. In short, frozen single-use vials of viable gonococci from characterized working banks containing the same lot of FA1090 A26 alone, FA1090Δ*mtrD* alone, or with mixtures of FA1090 A26 combined with FA1090Δ*mtrD* were thawed and sub-cultured overnight onto GCB plates (with supplements). Bacterial growth 18–24 h old consistent with *N. gonorrhoeae* morphology and with morphology that indicated piliation and predominantly Opa-negative colonies was harvested from culture plates with sterile swabs in 6 mL of sterile saline in sterile glass tubes. Optical densities were measured to determine and achieve the desired inoculum concentration of 4 x $10^6$ gonococci/mL, with further dilutions with sterile saline, if needed. For any given cohort of study participants, one single inoculum bacterial suspension was used for all individuals inoculated in that cohort within 15 minutes of preparation. On average, 0.25mL of *N. gonorrhoeae* inoculum was successfully delivered to the anterior urethra equating to approximately 1 million (1 x $10^6$) organisms, with established procedures as described previously [31,32,63].

Participants were observed for up to 5 days post-inoculation and received antibiotics as soon as discharge developed or on the final day of study (day 5 post-inoculation, irrespective of whether they were deemed to be infected or not). Antibiotic treatment was with cefixime (single 400 mg oral dose) or ceftriaxone (single 250 mg dose delivered intramuscularly). Infection alone without discharge did not trigger treatment. First-void urine specimens were obtained daily after inoculation for up to five days and examined quantitatively for pyuria (white blood cells) and gonococci (bacteriuria). Participants were considered to be infected if they had a positive urine culture or positive swab culture, with or without symptoms (i.e., urethral discharge). Urine sediment was cultured quantitatively on GC agar with 3 g vancomycin, 12.5 units nystatin, and 5 g trimethoprim lactate/ml, which permits the growth of FA1090 and FA1090Δ*mtrD*. Inocula used in single and mixed strain human challenges were also plated quantitatively. Colonies were enumerated for bacterial quantitation. Up to 96 colonies per volunteer per culture day and up to 96 colonies from mixed inocula were picked and stored in freezing medium until strain determination by real-time PCR.

## Murine competitive infections

Groups of mice were inoculated with *N. gonorrhoeae* in the laboratory of Dr. Ann Jerse at Uniformed Services University, Bethesda, Maryland with established procedures [19,64].

Female BALB/c mice (6 to 8 weeks old) were treated with Premarin and antibiotics to increase susceptibility to *N. gonorrhoeae* as described previously [19,64]. Mice were vaginally inoculated with the following mixtures of bacteria freshly grown from single-use vials of frozen gonococcal stocks: FA1090 and FA1090Δ*mtrD*, FA19 and FA19Δ*mtrD*, or FA19 and FA19 *mtrD*::*kan*. Mice and men were inoculated with suspensions made from the same lot of FA1090 and FA1090Δ*mtrD*. Mixtures of FA19 and FA19Δ*mtrD* or FA19 and FA19 *mtrD*::*kan* were prepared on challenge day, as previously described [19]. Vaginal swabs were collected every other day for 7 days starting on day 1 post-inoculation and suspended in 100 μl GC broth. Vaginal swab suspensions from mice inoculated with FA1090 and FA1090Δ*mtrD* or FA19 and FA19Δ*mtrD* were quantitatively cultured for *N. gonorrhoeae* on GCB plates with Sm (100 μg /ml) plus 10 μg PMB/ml. Up to 48 colonies per mouse per culture day were picked and stored in freezing medium until shipment to University of North Carolina at Chapel Hill for strain determination by real-time PCR, as described below, except for one cohort of 4 mice inoculated with FA1090 and FA1090Δ*mtrD* (mice 1–4) for whom real-time PCR data was obtained from up to 96 single colonies picked from the final day of positive cultures (day 5 post-inoculation; all mice cleared by day 7 post-inoculation) only. Longitudinal data real-time PCR data from day 1, day 3 and day 5 (day 5 post-inoculation was the last culture positive day for mice 5–1, who cleared gonococci by day 7 post-inoculation) were available from the remaining 7 mice inoculated with FA1090 and FA1090Δ*mtrD* (mice 5–11). For competitive infections with FA19 and FA19 *mtrD*::*kan*, vaginal swab suspensions were cultured on GC agar with Sm (100 μg /ml) for total fu counts (wild-type and mutant) and GC agar with Sm (100 μg /ml) plus Km (50 μg/ml) to determine mutant fu counts.

## Strain determination in human and mouse competitive infections that contained clean deletion mutants

Up to 96 colonies per human male volunteer per culture day and 48–96 single colonies per mouse per culture day were individually tested for strain determination using a duplex Taqman real-time PCR assay (S1 Fig). Between 48–96 single colonies from the quantitative culture of each inoculum suspension used to inoculate cohorts of men and mice with mixtures containing clean deletion mutants were also screened by real-time PCR. In brief, frozen, viable material that had been picked and stored from each colony underwent sub-culture in sterile 96 well microtiter plates by adding viable gonococci to 75 μL of sterile GC broth (with supplements) using aseptic technique and a replica plater tool. Gonococci were expanded overnight (18-24h) at 35–37˚C in a humidified atmosphere containing 5–7% $CO_2$ in order to generate enough bacterial material for DNA isolation and subsequent strain detection by real time PCR.

Bacterial lysates were generated according to established procedures [65]. Resulting lysates were used as DNA template in the real-time assay. In brief, each well containing bacterial material expanded from a single colony was placed in 75 μL of sterile lysis buffer containing 2 mM EDTA, 50 mM Tris Cl at pH 8.5, and 1% Triton X-100 [65]. Cells were lysed in 96-well PCR plates using a thermocycler under the following cycling conditions: 15 min at 94˚C followed by incubation for 5 min at 25˚C. Control lysates from each individual strain were made and included as controls on each assay plate.

The simultaneous and specific detection of wild-type and mutant strains was possible for three reasons (S1 Fig): 1) the wild-type specific primers and probe were located within the *mtrD* gene, which is deleted in the mutant strains, and produce a 164bp product (note: the *mtrD* gene in FA1090 and FA19 have 100% genetic homology); 2) the primers and probe targeting the mutant strain were designed in a gap PCR fashion, whereby the forward and reverse primers flank the *mtrD* deletion and the probe spans either side of the deletion; only if a

deletion is present, can the real-time assay utilize these primers and probe to amplify and detect the HEX-labeled 221 bp product; and 3) the two probes were labeled with two different fluorophores with non-overlapping emission spectra (wild-type probe was labeled at the 5' end with the FAM fluorophore, which has an emission spectrum of 510–530 nm; and mutant probe was labeled at the 5' end with HEX fluorophore, which has an emission spectrum of 560–580 nm). Real-time assays were run in 384 well plates with five positive and five negative controls on each plate using a ViiA 7 Real-Time PCR System from Applied Biosystems and the following cycling conditions: 1 cycle at 95°C for 3 min, followed by 40 cycles at each of 95°C for 15 sec and 58°C for 45 seconds. Primer and probe sequences, and PCR recipes are provided in S1 Table. Only wells which gave a positive amplification signal for one, but not both targets, were included for enumeration and competitive index calculation.

### Competitive index calculations

For experimental murine and human infections, results for infected individuals were expressed as a competitive index (CI) using the equation: CI = mutant cfu (output)/wild-type cfu (output) ÷ mutant cfu (input)/wild-type cfu (input). Output refers to the number of wild-type and mutant cfu enumerated from cultures of urine sediment or from cultures of mouse vaginal swab suspension

Input refers to the number of cfu enumerated from culture of bacterial suspension used to inoculate that cohort/group of men or mice. Thus, CIs reported for each participant and mouse were calculated with the exact proportion of strains identified in the inoculum of the group/cohort they were a part of. For human and mouse competitive infections, the culture limit of detection was assigned as 1 cfu/total number of recovered cfu for that sample (e.g., 1/ 96 or 1/48, respectively). The competitive index can be equal to 1 if there is equal fitness between the two strains, greater than 1 (or greater than 0 when expressed on the logarithmic scale) if the mutant is favored or less than 1 (or less than 0 when expressed on the logarithmic scale) if the wild-type is favored.

The fitness of the FA1090Δ*mtrD* mutant relative to wild-type FA1090 in human competitive infections was evaluated by computing the CIs on the day of clinical urethritis when antibiotic treatment was given. The assumption was that by treatment day when clinical urethritis was apparent, the "winning" strain with a fitness competitive advantage had successfully established a site of infection by outcompeting the "losing" or less fit strain. Because the mouse is not a natural host for *N. gonorrhoeae*, mice typically clear the the colonizing bacteria within 5–7 days of challenge. The final determination regarding competitive fitness of strains in the mouse model of infection was made using CIs on the last day of positive cultures. The assumption is that the bacteria on the last day of bacterial cultures are the ones that were the strongest colonizers.

### Statistics

The evaluable population in non-competitive infections included participants who received a dose of *N. gonorrhoeae* within 1 $Log_{10}$ of the intended dose and reached an objective study endpoint (urethral discharge or day 5). The evaluable population in competitive infections included participants who received a dose of *N. gonorrhoeae* within 1 $Log_{10}$ of the intended dose, reached an objective study endpoint (urethral discharge or day 5) and had at least one positive urine culture.

Differences between groups were evaluated by Mann-Whitney rank-sum tests, 1-sided Fisher exact tests, or Kruskall-Wallis test, as appropriate. Competitive indices (CIs) during

human and mouse competitive infections were tested against the null hypothesis that the median equal zero using the one-sample Wilcoxon sign rank test and alpha of 0.05.

## Financial disclosure

The authors and study were supported by funding from the National Institutes of Health. Specifically, MMH and JAD were both supported through Grant Award Numbers U01AI114378 and U19AI113170 from the National Institute of Allergy and Infectious Diseases. AW was supported though Grant Award Number U01AI114378 by the National Institute of Allergy and Infectious Diseases and though Grant Award Numbers TL1TR002491 and UL1TR001111 from the National Center for Advancing Translational Sciences. AEJ was supported through Grant Award Number U19AI113170 from the National Institute of Allergy and Infectious Diseases. WMS was supported through Grant Award Numbers U19AI113170 and R01AI021150 from the National Institute of Allergy and Infectious Diseases. WMS is the recipient of a Senior Research Career Scientist Award from the Department of Veterans Affairs Medical Research Service. The funders did not play a role in the study design, data collection and analysis, decision to publish, or preparation of the manuscript.

## Supporting information

**S1 Fig. Duplex Taqman real-time PCR assay design.**
(DOCX)

**S1 Table. Primer and probe sequences, and PCR mixture recipe.**
(DOCX)

**S2 Table. Strain composition of the gonococcal population in mixed inocula and of gonococci recovered from first void urine of men from treatment day.**
(DOCX)

**S2 Fig.** *In vitro* **growth curve of FA1090 and FA1090Δ*mtrD* when grown side-by-side in liquid GCB broth, as evidenced by optical density measurements (OD550).**
(DOCX)

**S3 Table. Strain composition of the gonococcal population in mixed inocula given to mice and of gonococci recovered from mouse genital swab cultures from the final positive culture day.**
(DOCX)

**S4 Table. Strain composition by qPCR in mixed inocula used in mouse challenge studies with wild-type FA1090 and FA1090Δ*mtrD* and FA19 and FA19*mtrD* and of gonococci recovered from mouse genital swabs collected from each positive culture day.**
(DOCX)

**S5 Table.** *In vitro* **competitive growth culture of FA1090 and FA1090Δ*mtrD*.**
(DOCX)

**S6 Table. Strain composition in mixed inocula used in mouse challenge studies with wild-type FA19 and FA19 *mtrD*::Kan and of gonococci recovered from mouse genital swabs collected from each positive culture day.**
(DOCX)

## Acknowledgments

We extend our gratitude to the study participants. The UNC-Global Clinical Trials Unit/ DMID 09–0106 Study Team conducted excellent work with respect to the recruitment and clinical procedures for this study. James Anderson and Lorraine Balleta provided technical support for the microbiologic assessments. We are grateful to Holly Baughman and Jill El-Khorazaty from Emmes for their input.

## Author Contributions

**Conceptualization:** Ann E. Jerse, William M. Shafer, Marcia M. Hobbs, Joseph A. Duncan.

**Data curation:** Andreea Waltmann.

**Formal analysis:** Andreea Waltmann, Nancy Hua, Marcia M. Hobbs, Joseph A. Duncan.

**Funding acquisition:** Ann E. Jerse, William M. Shafer, Marcia M. Hobbs, Joseph A. Duncan.

**Investigation:** Andreea Waltmann, Jacqueline T. Balthazar, Afrin A. Begum, Ann E. Jerse, William M. Shafer, Marcia M. Hobbs, Joseph A. Duncan.

**Methodology:** Andreea Waltmann, Jacqueline T. Balthazar, Afrin A. Begum, Ann E. Jerse, William M. Shafer, Marcia M. Hobbs, Joseph A. Duncan.

**Resources:** Jacqueline T. Balthazar, Afrin A. Begum.

**Supervision:** Ann E. Jerse, William M. Shafer, Marcia M. Hobbs, Joseph A. Duncan.

**Validation:** Andreea Waltmann, William M. Shafer, Marcia M. Hobbs, Joseph A. Duncan.

**Visualization:** Andreea Waltmann, Ann E. Jerse, Marcia M. Hobbs, Joseph A. Duncan.

**Writing – original draft:** Andreea Waltmann, Ann E. Jerse, William M. Shafer, Marcia M. Hobbs, Joseph A. Duncan.

**Writing – review & editing:** Andreea Waltmann, Ann E. Jerse, William M. Shafer, Marcia M. Hobbs, Joseph A. Duncan.

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
