## [Decision Letter · Decision Letter 0]

22 May 2024

Dear professor Duncan,

Thank you very much for submitting your manuscript "Experimental genital tract infection demonstrates Neisseria gonorrhoeae MtrCDE efflux pump is not required for in vivo human infection and identifies gonococcal colonization bottleneck" for consideration at PLOS Pathogens. Your manuscript was reviewed by members of the editorial board and by three independent reviewers. Considering the reviews (below), we would like to invite the resubmission of a significantly revised version that addresses the reviewers' comments, especially the Major Issues indicated by Reviewers 2 and 3, including a more thorough experimental assessment of the bottleneck in the murine model. In addition, all the comments in the Summary, Major Issues and Minor Issues from all three reviewers must be adequately addressed. 

We cannot make any decision about publication until we have evaluated the revised manuscript and your response to the reviewers' comments. Your revised manuscript is also likely to be sent to reviewers for further evaluation.

Sincerely,

Andreas J Baumler

Academic Editor

PLOS Pathogens

D. Scott Samuels

Section Editor

PLOS Pathogens

Michael Malim

Editor-in-Chief

PLOS Pathogens

orcid.org/0000-0002-7699-2064

Reviewer's Responses to Questions

**Part I - Summary**

Reviewer #1: In this manuscript, Waltmann et al. examines the role of the MtrCDE efflux pump of Neisseria gonorrhoeae in human and murine models of experimental infection. They show that the efflux pump is not required for infection of men in a strain that is not a strong expressor of the pump, provide evidence that a bottleneck exists during infection of the male urethra, and show strain differences (in an expressor vs. an under-expressor) in the requirement for expression of MtrCDE in female mice. As such, the data is novel and important, especially the evidence for a bottleneck during infection of male volunteers.

Reviewer #2: The initial aim of this manuscript is to assess the potential in vivo role of the MtrCDE efflux pump using the human challenge model. Using FA1090 the authors were not able to show a difference between the WT and mutant strain and concluded that in this model the efflux pump was not responsible for a competitive advantage. On the other hand, from their data they identified a bottleneck for GC infection. Furthermore as these results were apparently contradictory to previously published data showing a role of this locus using another strain, the authors tested their strain in a mice model and obtained the same result as in the human challenge model, thus concluding that the effect of the importance of this efflux pump was strain dependent.

As a whole the manuscript is easy to read and the data well presented. The major flaws of this manuscript which I would like to stress have already been discussed by the authors and unfortunately due to the use of the human model and the subsequent regulatory constraints cannot be experimentally address. The use of the strain FA10990 which is unable to upregulate the expression of the mtrCDE locus due to a spontaneous deletion in MtrA This is considerably reducing the message of this manuscript.

Reviewer #3: The authors present a comparative study that explores the behavior of N. gonorrhoeae in competitive index assays in men and laboratory mice. The influence on in vivo fitness of mtrD locus mutations in two gonococcal strain backgrounds was evaluated in mice; one strain background was evaluated in men. Valuable controls in men included mono-inoculation studies that helped determine that strain mixtures used in competitive index assays did not vary greatly in infectivity, inoculum size or pyuria. The bulk of the manuscript focused on competitive index assay data. Unexpectedly, mtrD mutations in the FA 1090 background did not impact fitness in the human challenge model of urethra infection. This differs from results in a murine model of vaginal infection where mtrD mutations impair FA19 strain fitness in vivo.

This study builds on decades of previous work and is to be commended for attempting to merge human and murine infection models’ ability to inform knowledge of host-bacteria interactions that can help in design of vaccines and therapies.

Competitive Index assay data would foster additional confidence if it was presented in a longitudinal format for the human and mouse studies. It was not clear why the authors only reported competitive index data for “final positive culture days” in humans and mice. Longitudinal data is presented in Figure 2B for the male volunteers but it would be nice to see (where possible) the competitive index data presented longitudinally for days post-inoculation using the format of data in Figure 1. This is a typical way to present competitive index data in at least one reference cited by the authors (such as reference 8, PMID: 18761689).

The authors acknowledge that their data could be influenced by multiple factors including model systems, strain backgrounds and how a mutant strain was generated. I was not fully convinced that other variables also contributed to some of the differences noted. A more compelling presentation of bottleneck effects in the FA 1090 murine model would increase my enthusiasm for the study.

**Part II – Major Issues: Key Experiments Required for Acceptance**

Reviewer #1: None

Reviewer #2: As mentioned by the authors a bottleneck is identified using two derivatives of the same strain with identical fitness. In this manuscript the authors aimed at assessing whether there is a different of fitness and concluded with a strain unable to fully expressed the WT phenotype that there is no competitive advantage of the locus under study but on the other hand identified a bottleneck. From the experimental design I have to admit I had to take some time to convince myself that this conclusion was correct. However, from the data with FA1090 in mice, this bottleneck does not seem completely as obvious in the mice model. It would be very nice if the authors could at least clearly identify this bottleneck in the mice model using two strains with the very same fitness.

Reviewer #3: Questions and comments.

1. Regarding the inoculum descriptions such as those presented in Supplementary tables 2 and 3:

a. Was each group inoculated with the same lot of inoculum? It was not clear if the each group received the same inoculum lot from the same frozen stock. It was not clear if the reported mtrD and WT strain percentages in the inoculum show variation within the same inoculum lot or variation between different lots of inoculum.

b. Was the age of the frozen inoculum or inocula used for the male volunteer studies between April and November of 2018 quite variable? It was not clear if study participants or mice received inoculum or inocula of widely different ages (stored at -70C for widely different amounts of time).

c. Was the % of each strain in the strain mixtures quantified every time a stock was thawed prior to an inoculation?

2. Were any of the same lots of inoculum used for inoculation of men used in mice or were they completely different lots? If different lots were used were they prepared in the same lab from the same founder stocks? This type of information in the methods section would be appreciated.

3. Was the in vitro fitness of the strains created for this study analyzed? If not, should this be mentioned in the manuscript? If yes, could the authors describe the in vitro characterization?

4. On lines 227-228 it is mentioned that CFUs were measured up to five days after inoculation. Why is the competitive index data not reported for the earlier time points? The methods section reports that swabs were collected every other day but only data from the final sampling is reported. It would build confidence in the study if similar trends in the competitive index data were found to occur at multiple time points post-inoculation.

5. Why were there not supplementary tables provided to show the strain percentages in strain mixture inocula used for FA19 studies? It seems like that would help the authors compare if PCR and culture-based measurements of competitive index yield similar data.

6. For figure 2, could the authors add a panel C to present similar data from the FA 1090 infected mice over time? It seems like the authors are contending the FA 1090 data supports the existence of a bottle neck in man and mice.

7. I feel the 60% mutant/40% wild type pie chart in Figure 2B is slightly misleading. Would the authors consider adding pie charts for the inoculum each participant received on the left side of the panel (using the numbers from supplementary table 2)? This would help readers identify which participants received the same or similar inoculum. It might also be worthwhile to add the group number in parentheses behind each participant number so readers can correlate participants with information provided in supplementary table 2.

8. The participants’ pie chart data in Fig. 2B appears to have been organized in a manner that resembles option 4 in Figure 2A. It is not organized according to group as in the supplementary table 2. I found Figure 2A a bit confusing because for options 2 and 3 only “Wild-type advantaged” was depicted. It seems like those options could be divided into parts A and B and you could depict both “Wild-type advantaged” and “mutant advantaged”.

9. Lines 414-416. My reading of your manuscript led me to conclude you think there is a bottle neck for FA 1090 in the murine model too. Why can’t you add your mouse data as a panel C to Figure 2 to show how it compares to the human data?

10. Is one way to interpret the data in Figure 2B that the mtrD mutants are capable of causing longer periods of asymptomatic urethritis? It seems like participants dominated by wild-type FA 1090 exited the study earlier (I assume - due to detection of urethritis and treatment). While fitness for both strains was not significantly different it seems like the two strains differ in their asymptomatic persistence and timing of urethritis symptoms.

11. Line 398. The references cited for single-cell bottlenecks are all quite dated. Tn-seq has been very helpful in elucidating bottlenecks for infectious bacteria in animal models. Many human bacterial pathogens have much larger bottle necks than single cell bottlenecks. I don’t find the bottleneck discussion very compelling and think it needs updated with more current examples from the literature.

12. Is it worthwhile to add supplementary graphs showing longitudinal CFUs/ml of urine for the human participants and CFUs/vaginal swab for each mouse. It might be another way of illustrating CFU burden differences between the two models.

13. Figure 1 could be improved if the symbols could be annotated with the day post-inoculation the data came from. Could a superscript be added to each symbol for the day post-inoculation the data came from? Alternatively, could different colored borders be added to each symbol and a legend added indicating the day post-inoculation the competitive index data came from.

**Part III – Minor Issues: Editorial and Data Presentation Modifications**

Reviewer #1: 1. General. It is usual practice to state the age (range, mean + SD) and ethnicity of the human volunteers who participated in the infection experiments. Please add that data to the text or Tables.

2. Line 147, correct “condidates”

3. Line 152, insert a comma before “and”

4. Line 161, complete the sentence “demonstrate a gonococcal infection bottleneck for the first.”

5. Line172, it is unclear whether FA1090 A26 (which lacks MtrA expression) is considered to be the “wild type” strain or is a derivative of FA1090. Please clarify. If you choose to use the A26 designation, please use it throughout (for example, FA1090 is used in Table 2, not FA1090 A26). I happen to know that the A26 designation refers to a stock of FA1090 that is used for human challenges due to its defined pilation and opa states, but the general reader will not know this, and we don’t find that out until line 447 in the methods.

6. Line 194-195 , change “and thus processed urines were not the first void of the day” to “of providing first voided urine.”

7. Line 232. Please address discrepancies between text, Figure 1B and Supplementary Table In text, it is stated “we observed five outcomes that favored the wild type FA1090 and six outcomes that favored FA1090ΔmtrD, but the supplementary table indicates there were only 10 evaluable mice, not 11, with 5 outcomes that favored the wild type and 5 outcomes that favored the mutant. Figure 1B also has 11 outcomes.

8. Figure 1 text states it is using horizonal bars for medians of each group, but it also appears to be using the same horizontal bars for the CIs (not stated in the legend); use of the same symbol for different purposes is confusing.

9. Fig 2B. Please move the pie chart showing the inoculum to the left of the Figure panel (currently it is within the panel and aligns with volunteer 313)

10. Line 290 is this a polar mutant?

11. The first paragraph of the discussion is nearly 50 lines long. Please consider a paragraph break after you finish discussing the murine data in line 311 and have a separate paragraph about the human data.

12. Line 384, 385, 415, 425 fix “lackthereof” , “variants variants” "porentially", “sexually-transmitetd”; please re-check for typos throughout

13. Lines 418-23 the statement “that by showing gonococci can initiate productive infection in the human male urethra even when the pump is not functional, we have generated data that can inform future gonorrhea vaccine and drug development efforts that target the function of the Mtr efflux pump” is vague. Please expand the discussion to state whether the Mtr pump remains a reasonable drug or vaccine target, given all the human data you present here. It would also be better if the last paragraph of the discussion were split into 2 parts, one focused on the pump and the last focused on the bottleneck, which deserves to stand on its own, as that may be the most important finding here.

Reviewer #2: The last part of the discussion is not really necessary.

Reviewer #3: Line 147. Correct spelling of “condidates”.

Line 218. “two models” – I assume this refers to the human challenge model and the BALB/c mouse model. It is easy to get lost and might help readers like myself to edit “two models” to “human and murine models showed similar results”.

Line 424. Correct spelling error – “transmitetd”.

Line 594. The timing of the human infections is listed but not the murine experiments.

Line 689. It would be helpful to adjust the legend or panel 1A so it was clear that the numbers in panel 1A refer to “infection outcomes”.

Lines 625, 643. “wbc” not defined.

“Tx” not defined in supplementary tables.

Fig. 1B “FA 1090” is missing from the upper text on the right side of the graph.

I wish the authors could add something to their discussion about “transmission restriction” mentioned on line 162.

PLOS authors have the option to publish the peer review history of their article (what does this mean?). If published, this will include your full peer review and any attached files.

Reviewer #1: No

Reviewer #2: No

Reviewer #3: **Yes: **Nathan John Weyand
---

## [Editor Report · Decision Letter 1]

10 Sep 2024

Dear professor Duncan,

We are pleased to inform you that your manuscript 'Experimental genital tract infection demonstrates Neisseria gonorrhoeae MtrCDE efflux pump is not required for in vivo human infection and identifies gonococcal colonization bottleneck' has been provisionally accepted for publication in PLOS Pathogens.

Best regards,

Andreas J Baumler

Academic Editor

PLOS Pathogens

D. Scott Samuels

Section Editor

PLOS Pathogens

Michael Malim

Editor-in-Chief

PLOS Pathogens

orcid.org/0000-0002-7699-2064

---

## [Editor Report · Acceptance letter]

20 Sep 2024

Dear professor Duncan,

We are delighted to inform you that your manuscript, "Experimental genital tract infection demonstrates Neisseria gonorrhoeae MtrCDE efflux pump is not required for in vivo human infection and identifies gonococcal colonization bottleneck," has been formally accepted for publication in PLOS Pathogens.

Best regards,

Michael Malim

Editor-in-Chief

PLOS Pathogens

orcid.org/0000-0002-7699-2064